# Neuronal Dynamics and miRNA Signaling Differ between SH-SY5Y *APPSwe* and *PSEN1* Mutant iPSC-Derived AD Models upon Modulation with miR-124 Mimic and Inhibitor

**DOI:** 10.3390/cells10092424

**Published:** 2021-09-14

**Authors:** Gonçalo Garcia, Sara Pinto, Mar Cunha, Adelaide Fernandes, Jari Koistinaho, Dora Brites

**Affiliations:** 1Neuroinflammation, Signaling and Neuroregeneration Laboratory, Research Institute for Medicines (iMed.ULisboa), Faculty of Pharmacy, Universidade de Lisboa, 1649-003 Lisboa, Portugal; ggarcia@campus.ul.pt (G.G.); sfcpinto@campus.ul.pt (S.P.); mariana.xcf@gmail.com (M.C.); 2Department of Pharmaceutical Sciences and Medicines, Faculty of Pharmacy, Universidade de Lisboa, 1649-003 Lisboa, Portugal; amaf@ff.ulisboa.pt; 3Instituto de Medicina Molecular, Faculdade de Medicina, Universidade de Lisboa, 1649-003 Lisboa, Portugal; 4Central Nervous System, Blood and Peripheral Inflammation, Research Institute for Medicines (iMed.ULisboa), Faculty of Pharmacy, Universidade de Lisboa, 1649-003 Lisboa, Portugal; 5A.I. Virtanen Institute for Molecular Sciences, University of Eastern Finland, FI-70211 Kuopio, Finland; jari.koistinaho@uef.fi or; 6Neuroscience Center, Helsinki Institute of Life Science (HiLIFE), University of Helsinki, FI-00014 Helsinki, Finland

**Keywords:** Alzheimer’s disease (AD), cell experimental models, inflammatory-associated miRNAs, iPSC-derived neurons, miR-124-3p modulation, neuronal dysfunction, neuropathological hallmarks of AD, paracrine signaling, secretome, small extracellular vesicles (exosomes)

## Abstract

Neuronal miRNA dysregulation may have a role in the pathophysiology of Alzheimer’s disease (AD). miRNA(miR)-124 is largely abundant and a critical player in many neuronal functions. However, the lack of models reliably recapitulating AD pathophysiology hampers our understanding of miR-124’s role in the disease. Using the classical human SH-SY5Y-*APP695 Swedish* neuroblastoma cells (SH-*SWE*) and the *PSEN1* mutant iPSC-derived neurons (iNEU-*PSEN*), we observed a sustained upregulation of miR-124/miR-125b/miR-21, but only miR-124 was consistently shuttled into their exosomes. The miR-124 mimic reduced *APP* gene expression in both AD models. While miR-124 mimic in SH-*SWE* neurons led to neurite outgrowth, mitochondria activation and small Aβ oligomer reduction, in iNEU-*PSEN* cells it diminished Tau phosphorylation, whereas miR-124 inhibitor decreased dendritic spine density. In exosomes, cellular transfection with the mimic predominantly downregulated miR-125b/miR-21/miR-146a/miR-155. The miR-124 inhibitor upregulated miR-146a in the two experimental cell models, while it led to distinct miRNA signatures in cells and exosomes. In sum, though miR-124 function may be dependent on the neuronal AD model, data indicate that keeping miR-124 level strictly controlled is crucial for proper neuronal function. Moreover, the iNEU-*PSEN* cellular model stands out as a useful tool for AD mechanistic studies and perhaps for the development of personalized therapeutic strategies.

## 1. Introduction

Neuronal miRNA (miR)-124 is one of the most abundant miRNAs in the brain [1]. It was firstly shown to control the choice between neuronal and astrocyte differentiation [2], as well as to promote neuronal differentiation [3] and neurite outgrowth [4]. However, electrophysiological studies showed that miR-124 also constrains synaptic plasticity [5]. Though not found in macrophages, miR-124 is expressed in microglia, where it was associated to their deactivation and suppression of experimental autoimmune encephalomyelitis [6]. Interestingly, miR-124 was referred to ameliorate the motor symptoms in Parkinson’s disease and to have pro-neurogenic potential when loaded in polymeric nanoparticles [7]. By using SH-SY5Y neuroblastoma cells, An et al. demonstrated that miR-124 is a potent negative regulator of β-site amyloid precursor protein-cleaving enzyme (BACE1) and that it may be involved in Alzheimer’s disease (AD) pathology, considering its reduced expression in AD sporadic patients [8]. In contrast, a marked miR-124 upregulation in the hippocampus and temporal cortex was observed in AD patients and in the Tg2576 transgenic mice, where it was associated to an elevation of the amyloid-β (Aβ) peptide 1–42 [9]. These authors identified the miR-124/PTPN1 pathway as a critical mediator of synaptic dysfunction and memory loss in AD. Therefore, the results about the beneficial or harmful effects of miR-124 expression levels in AD, which may result from the different models, mutations or disease stages used in such studies, are controversial. When we assessed miR-124 expression in human hippocampal homogenates from healthy individuals and from AD patients in Braak stages II, III, IV and V/VI, we only observed a significant elevation of miR-124 in stage III [10]. Such a finding may relate to a potential defensive mechanism that is induced at disease middle stages and later disappears due to neuronal demise. In contrast, the role of miR-124 in amyotrophic lateral sclerosis (ALS) seems to be linked to pathological processes. In conformity, we found that miR-124 elevation in the mutated SOD1 (mSOD1) motor neurons was associated to neurodegeneration and paracrine signaling dysregulation in N9 microglia and spinal organotypic cultures from the transgenic mSOD1 mice [11]. Besides the mitochondrial accumulation of TAR DNA-binding protein 43 (TDP-43) that may occur in AD and ALS [12], the diseases have different pathological signatures. AD affects multiple brain regions critical to learning and memory and the most common cellular and molecular hallmarks are extracellular Aβ plaques and neurofibrillary tangles of hyperphosphorylated Tau protein [13]. As for ALS, the disease specifically affects upper and lower motor neurons leading to neuromuscular junction instability and distal axonal degeneration [14].

Using an in vitro model of AD, we previously observed an elevation of miR-124 in the dopaminergic SH-SY5Y neuroblastoma cell line expressing the *APP695 Swedish* mutant gene (SH-*SWE*), as compared to their naïve SH-SY5Y (SH-*WT*) cells [15]. These cells that mimic the usual neuronal morphology and neurite outgrowth showed, after differentiation with the retinoic acid (RA) [16], an overexpression of several reactive (e.g., S100B) and inflammatory transcripts (e.g., HMGB1, TNF-α), as well as the release of Aβ1-40 when expressing the mutated *APP* [15]. However, we must also consider that RA is known to affect cell susceptibility to neurotoxins by inducing higher immunologic tolerance and lower neurotoxic responses [17]. In all ways, the expression of a variety of markers of mature neurons makes SH-SY5Y cells useful to investigate several diseases, including AD, and to obtain accurate results that may be translated into in vivo models [18]. In fact, the evaluation of RA-maturated SH-SY5Y cells by proteomic analysis revealed upregulated proteins that were involved in neuritogenesis [19]. Transfection with the *APP695 Swedish* mutant gene did not affect cell morphology and resulted in neurotoxicity [20]. The model is then considered useful to study AD pathogenesis [21] and to test novel therapeutic strategies [22], but the modeling of AD with induced pluripotent stem cell (iPSC)-derived neurons (iNeurons) brought new potentials in the recapitulation of molecular mechanisms of AD. Indeed, iNeurons better mimic the in vivo conditions and hold great promise to assess neurotoxic mechanisms, though they show differences in spontaneous neuronal activity [23,24,25]. These cells have even been proposed for AD treatment by transplantation [26]. A study using iPSC lines generated from familiar AD patients revealed increased Aβ42:40 in all, but quantitatively and qualitatively divergent Aβ secretomes, and that distinct presenilin 1 (*PSEN1*) mutations caused different alterations, altogether culminating in the reduction in γ-secretase carboxypeptidase-like activity [27]. Dysregulation of endocytosis-associated genes and early endosome enlargement were observed in human iPSC lines carrying *APP* and/or *PSEN1* mutations [28]. Nonetheless, experimental variations between studies associated to cellular heterogeneities, as well as to transcript and protein abundance, may occur with iPSCs, and the maturity of the generated iNeurons may fail in reproducing the biological age at disease onset [29]. Interestingly, despite the evidence that miR-124 induces neuronal differentiation and functional maturation [30], to the best of our knowledge there are no studies addressing the expression levels of miR-124 in iNeurons from AD iPSCs.

Considering that experimental models of AD are critical to advance knowledge on the underlying pathological mechanisms and that all have limitations [31], we here used two different approaches to gain insight into the potential contribution of neuronal miR-124 expression levels in AD-associated pathological mechanisms. We used the classical SH-*WT* and the SH-*SWE* cell lines based on our previous experience with these models [15], and the advanced model of dopaminergic midbrain iNeurons generated from iPSCs derived from a healthy individual (iNEU-*WT*) and a patient carrying a deletion in the exon 9 of the *PSEN1* gene (iNEU-*PSEN*). The original iPSCs were already characterized, described, and used after differentiation into astrocytes to show associated pathological features [32], and into microglia to evidence a disparate phenotype relatively to those obtained from patients with the APP-*SWE* mutation and with the apolipoprotein E4 (APOE4) variant [33]. Therefore, considering the controversy around the role that miR-124 may have on the AD onset and progression and the unique opportunity to have these two different AD experimental models, we decided to transfect both cell lines with either the miR-124 inhibitor or the miR-124 mimic to reduce and increase its expression levels, respectively, and to investigate the consequences on neuronal dynamics, predictive function and paracrine signaling.

We first assessed the basal miR-124 expression levels, together with other miRNAs associated to neuroinflammation (inflamma-miRNAs), in such different cell experimental models and in their small extracellular vesicles that will be designed here as exosomes. Mature miRNAs are derived from a duplex precursor, where the -5p strand or “guide” is preferentially incorporated into the RNA-induced silencing complex and the complementary -3p strand or “passenger” is believed to be more prone to being degraded [10]. Here, we evaluated the following specific strands: miR-124-3p (the most assessed in the literature), miR-21-5p, miR-125b-5p, miR-155-5p and miR-146a-5p. They will be used from now on without specifying the strand unless required. Consequences of miR-124 up- and downregulation were evaluated on dendrite outgrowth and mitochondria dynamics, spine density, APP and Aβ processing, Tau phosphorylation and miRNA content in the secretome (either as free species or as part of exosomal cargo).

Data show the existence of similar and dissimilar pathological events in the two AD neuronal models used, which help to further understand the AD-associated disparate results on miR-124 in the literature. Globally, we provide evidence that miR-124 upregulation is associated to the prevention of some AD-associated pathological mechanisms and the propagation of inflamma-miRNAs mediated by exosomes, once such effects disappeared by miR-124 downregulation. Our results also highlight the importance of stratifying patients toward more effective and patient-centered care using miR-124 as a biomarker and therapeutic target in AD pathology.

## 2. Materials and Methods

### 2.1. Culture and Differentiation of Human SH-WT and SH-SWE Cells

SH-*WT* (naïve) and SH-*SWE* (transfected with human *APP695*) cells were a gift from Professor Anthony Turner. Cells that were routinely tested for mycoplasma contamination were cultured in Dulbecco’s Modified Eagle’s Medium (DMEM) (Gibco, Thermo Fisher Scientific, Waltham, MA, USA), supplemented with 10% Fetal Bovine Serum (FBS) and 2% AB/AM, in T75 flasks. Briefly, cells were maintained in a humidified atmosphere at 5% CO_2_, 37 °C, and seeded into 12-well plates coated with poly-D-lysine (100 µg/mL, Sigma-Aldrich, St. Louis, MO, USA) and laminin (4 µg/mL, Gibco, Thermo Fisher Scientific, Waltham, MA, USA) at a final concentration of 5 × 10^4^ cells per well. To induce neuronal differentiation, we used 10 μM of RA and the medium was changed every day for 7 days, as previously described [34] and currently implemented in our laboratory [15].

### 2.2. Culture of Human iPSCs, Neuronal Differentiation and Cell Maturation

iPSCs, generated from a female AD patient carrying *PSEN1* exon 9 deletion (*PSEN1ΔE9* mutation) and from a healthy female control, were kindly provided by Dr J. Koistinaho (from the University of Helsinki; in the past, from the University of Eastern Finland), under the JPco-fuND 2015 project MADGIC, covered by Material Transfer Agreement, and in close collaboration with the project team partners Dr F. Edenhofer (from the University of Innsbruck) and Dr T. Malm (from the University of Eastern Finland). iPSC lines were generated after approval of the committee on Research Ethics of Northern Savo Hospital district (123/2016) and written consent from the subjects. Demographic information of each iPSC-line is summarized in Appendix A. Full characterization in terms of genetic stability and pluripotency markers of such iPSCs was previously published in studies using their differentiation into astrocytes (iAstrocytes) and microglia (iMicroglia) for characterization of AD-associated glial aberrancies [32,33]. In our laboratory, iPSCs were differentiated into the midbrain dopaminergic neurons iNEU-*WT* (from a healthy female control) and iNEU-*PSEN* (from a female pre-symptomatic AD patient), following described and established protocols [32,35], with minor modifications, as schematized in Appendix A. In brief, iPSCs were grown on Matrigel-coated (Corning, Corning, NY, USA) plates in Essential 8 (E8) medium and passaged with 0.5 mM EDTA. Freshly passaged cells were cultured with 5 mM Y-27632 ROCK inhibitor (Selleckchem, Houston, TX, USA). Differentiation started by changing to neural differentiation medium (NDM) consisting of DMEM/F12 and Neurobasal (1:1), 1% B27 without vitamin A, 0.5% N2, 1% Glutamax, and 0.5% penicillin/streptomycin (50 IU/50 mg/mL) (all from Invitrogen, Waltham, MA, USA), supplemented with dual SMAD inhibitors 10 mM SB431542 (Sigma) and 200 nM LDN193189 (Selleckchem). Medium was changed daily for 12 days, until rosette-like structures started to emerge (Appendix A). Cells were then cultured in NDM supplemented with 20 ng/mL basic fibroblast growth factor (bFGF) for 2–3 days to expand the rosettes. Areas with rosettes were mechanically lifted and cultured in suspension on ultra-low attachment plates (Corning) in NDM for 2 days to allow for neural progenitor sphere (NPS) formation. Then, spheres were maintained in NDM supplemented with 10 ng/mL bFGF and 10 ng/mL epidermal growth factor (EGF) (Peprotech, Rocky Hill, NJ, USA) for 1 month. Medium was changed every 2–3 days and spheres were split manually every week. For neuronal maturation, spheres were dissociated with Accutase (STEMCELL Technologies, Vancouver, BC, Canada) and plated on Poly-L-ornithine (20 μg/mL) + Laminin (10 μg/mL)-coated dishes and differentiated using Neuron Induction Medium (NIM) consisting of DMEM-F12 and Neurobasal (1:1), 1% N2, 2% B27 without vitamin A, 1% L-Glutamax, and 0.5% penicillin/streptomycin (50 IU/50 mg/mL) (all from Invitrogen) supplemented with 20 ng/mL of neuronal BDNF and GDNF growth factors (Peprotech), during 1 month prior to experiments. Neuronal differentiation and maturation were confirmed by monitoring Tau, microtubule-associated protein MAP-2 and F-actin immunostaining at week 11 (Appendix A), followed by a deep look at synaptic dynamics by the immunofluorescence of pre-synaptic SV-2 and synapsin-1 markers, together with the post-synaptic PSD-95 protein.

### 2.3. Isolation of Exosomes from the Cell Secretome

Exosomes were isolated by differential ultracentrifugation from the cell secretomes, as we described [15,36]. Secretomes were obtained from the differentiated SH-*WT*/SH-*SWE* cells and iNEU-*WT*/iNEU-*PSEN* cells, either untreated or after miR-124 modulation, as described in Section 2.7. Equal volumes of cell media were promptly centrifuged at 1000× *g* for 10 min, to pellet cell debris. The supernatants were transferred into new tubes and centrifuged at 16,000× *g* for 1 h, to pellet and discard the large extracellular vesicles, usually denominated as microvesicles. The remaining supernatant was filtered using a 0.22 μm pore size membrane and centrifuged at 100,000× *g* for 2 h, using the Ultra L-XP100 centrifuge (Beckman Coulter, Brea, CA, USA) to isolate the exosomes. The pellet was resuspended/washed in phosphate-buffered saline (PBS) and centrifuged once again at 100,000× *g* for 2 h. To evaluate miRNA content, pellets were then suspended in 200 μL lysis buffer for RNA extraction with the miRCURY Isolation Kit-Cell (Exiqon, Gill StreetWoburn, MA, USA). For protein extraction, pellets were resuspended in 50 µL Cell Lysis buffer (Cell Signaling, Danvers, MA, USA), transferred into microtubes, snap-frozen and stored at −80 °C until further quantification/analysis. For Transmission Electron Microscopy (TEM), freshly isolated exosomes were resuspended and kept in ice-cold PBS for 1–2 days until analysis.

### 2.4. Characterization of Exosomes for Morphology, Protein Markers, Concentration and Size Distribution

Exosome morphology was assessed by TEM, using equal volumes of exosome suspensions that were dried onto freshly ‘glow discharged’ 300 mesh formvar/carbon-coated TEM grids (Ted Pella, Redding, CA, USA), negatively stained with 2% aqueous uracyl acetate and observed under a TEM with a JEOL JEM 1400 microscope (JEOL Ltd., Tokyo, Japan) at an accelerating voltage of 120 kV. Images were digitally recorded using a Gatan SC 1100 ORIUS CCD camera (Gatan Inc., Warrendale, PA, USA). Round cup-shaped structures, ranging from 50 to 200 nm size were considered exosomes. Exosomes were also characterized by the presence of ALIX, CD63 and flotillin proteins that are usually present in these small extracellular vesicles. The presence of such exosomal markers was evaluated by Western blot. Exosome samples were processed for protein extraction and quantification, as described below. Since differences in the total protein content were detected in samples collected from different cell models (Appendix A), the same protein amount (30 µg) was used in every condition.

Concentration and size of exosomes were evaluated by Nanoparticle tracking assay (NTA) using the Nanosight (model LM10-HSBF, Malvern Instruments, Malvern, UK). Samples were injected into the system under controlled flow using a NanoSight syringe pump and integrated scripting control system. Five different videos up to 60 s long were made and particle movement was analyzed by NTA-software (version 3.1).

### 2.5. Protein Quantification and Western Blot

Concentration of proteins was determined by using the BCA Protein Assay Kit (Pierce, Biotechnology, Waltham, MA, USA). Samples (30 µg of exosomes, 40 µg of SH-*WT* or SH-*SWE* cells and 20 µg of iNEU-*WT* and iNEU-*PSEN* cells) were separated on Tris-Tricine gel, transferred into nitrocellulose membranes (Amersham, Health, Buckinghamshire, United Kingdom) and incubated in blocking buffer (5% (*w*/*v*) non-fat dried milk in Tween 20 (0.1%) tween-tris buffer saline (T-TBS)) for 1 h at room temperature. Membranes were incubated at 4 °C overnight with primary antibodies: mouse anti-ALIX (1:1000, Cell Signaling); goat anti-CD63 (1:1000, Santa Cruz Biotechnology, Dallas, TX, USA); mouse anti-flotilin-1 (1:1000, BD Biosciences, Franklin Lakes, NJ, USA); mouse anti-amyloid β (1:1000, Merck Millipore, Burlington, MA, USA); rabbit anti-phospho-Tau (Ser404) (1:1000, Cell Signaling); rabbit anti-Tau (1:1000, Synaptic Systems, Goettingen, Germany); and mouse anti-β-actin (1:2000, Sigma Aldrich, St. Louis, MO, USA). Each antibody was diluted in blocking buffer, followed by incubation with the respective secondary antibodies for 1 h at room temperature, using HRP-conjugated goat anti-mouse (1:2000), rabbit anti-goat (1:2000), and goat-anti-rabbit (1:2000), all from Santa Cruz Biotechnology. WesternBright Sirius (Advansta, San Jose, CA, USA) was used as the chemiluminescent substrate and the densitometric analysis of protein bands obtained with the ChemiDoc Imaging System (Bio-Rad, Hercules, CA, USA). Relative intensity of protein bands was estimated using the Bio-Rad Image Lab analysis software (Bio-Rad).

### 2.6. RNA Extraction and RT-qPCR

Total RNA was extracted from differentiated SH-*WT*/SH-*SWE* and iNEU-*WT*/iNEU-*PSEN* cells, using TRIzol^®^ reagent (LifeTechnologies), according to manufacturer’s instructions. Total RNA (enriched in miRNAs) from the exosomes was extracted using the miRCURY^TM^ LNA^TM^ Universal RT miRNA PCR kit (Qiagen, Venlo, The Netherlands). Quantification was performed with Nanodrop^®^ ND-100 Spectrophotometer (NanoDrop Technologies, Wilmington, DE, USA). For miRNA determination, equal amounts of RNA were reverse transcribed into cDNA using the Universal cDNA Synthesis Kit (Qiagen). Then, miRNA expression was determined by Real-Time Quantitative Polymerase Chain Reaction (RT-qPCR) using the miRCURY LNA^TM^ Universal RT miRNA PCR kit (Qiagen). For mRNA determination, same amounts of total RNA were reverse transcribed into cDNA using the GRS cDNA Synthesis Master Mix kit (GRiSP, Porto, Portugal) and RT-qPCR was performed using Xpert Fast Sybr Blue (GRiSP) as master mix with specific predesigned primers (Appendix A). Both miRNA and mRNA RT-qPCR were run on QuantStudio 7 Flex RT-PCR System (Applied Biosystems, Waltham, MA, USA). Running conditions for miRNAs consisted of polymerase activation/denaturation and well-factor determination at 95 °C for 10 min, followed by 50 amplification cycles at 95 °C for 10 s and 60 °C for 1 min (ramp-rate 1.6 °C/s). Running conditions for mRNA determination were 50 °C for 2 min followed by 95 °C for 2 min, 40 cycles at 95 °C for 5 s and 62 °C for 30 s. Melt-curve analysis was performed after amplification, and the specificity of PCR products was confirmed. Expression data of at least four independent experiments were processed using the 2^−ΔΔCT^ method with the internal control glyceraldehyde 3-phosphate dehydrogenase (GAPDH) for mRNA and that of U6 for miRNA, together with the exogenous control Spike-in dataset. The results were expressed as fold change.

### 2.7. Modulation of miR-124 Levels in Neuroblastoma and iPSC-Derived Neurons

Differentiated SH-*WT*/SH-*SWE* cells and iNEU-*WT*/iNEU-*PSEN* cells were changed to Optimem medium (Gibco, Thermo Fisher Scientific, Waltham, MA, USA) and transfected with pre-miR-124-3p (mimic) and anti-miR-124-3p (inhibitor) (Ambion, Austin, TX, USA), each at 15 nM/well and cultured overnight. Mock transfected and negative controls (Scramble sequence provided by Qiagen) for both mimic and inhibitor were performed in parallel, using the same concentrations and conditions. Transfection agent (X-tremeGENE—Sigma Aldrich) was equally applied in each circumstance. Since we observed that mock and negative control produced very similar results, only mock control data will be presented in subsequent assays. SH-*SWE* cells were changed to FBS-free media, while iNEU-*PSEN* and iNEU-*WT* cells were cultured in Neurobasal/DMEM-F12 mixture (1:1). Cells were cultured for 24 h. At the end, the cell secretomes were collected for exosome isolation, characterization, and inflamma-miRNA determination. Cells were also processed for morphological and cellular dynamics, including miRNA content.

### 2.8. Evaluation of Cell Viability by the Nexin Assay

To determine the viability of adherent and floating cells, cells were collected from the culture medium or detached with trypsin, mixed, and spun down at 500 g for 5 min. Pellet was resuspended in 1% bovine serum albumin (BSA) in PBS and stained with phycoerythrin-conjugated annexin V (Annexin V-PE) and 7-amino-actinomycin D (7-AAD), using the Guava Nexin Reagent^®^ (Merck Millipore). Stained cells were analyzed with a flow cytometer (Guava easyCyte 5 HT, Merck Millipore), using the Guava Nexin Software. Four cellular populations were distinguished: viable cells (Annexin V-PE and 7-AAD double-negative), early apoptotic cells (Annexin V-PE positive and 7-AAD negative), late apoptotic cells (Annexin V-PE and 7-AAD double-positive) and necrotic cells/cellular debris (Annexin V-PE negative and 7-AAD positive).

### 2.9. Immunocytochemistry for Mitochondrial Fusion/Fission, Cytoskeletal and Synaptic Proteins

We evaluated the rates of mitochondrial fission and fusion by assessing the fission dynamin-related protein 1 (DRP1) and mitofusin-2 (MFN-2), respectively. Additionally, we determined the expression levels of synaptic proteins, e.g., the postsynaptic density protein 95 (PSD-95) and the synaptic vesicle protein SV2 (SV-2). Lastly, we evaluated the microtubule-associated proteins Tau and MAP-2, as well as the cytoskeleton-associated F-actin patterns in dendrites and axons.

Cells were plated onto coverslips and fixed with paraformaldehyde (4% *w*/*v* in PBS) for 20 min. Then, cells were washed with PBS and permeabilized with Triton-X100 0.2% in PBS for 10 min. Blocking was performed with BSA at 3% in PBS for 30 min. F-actin was stained using phalloidin conjugated with AlexaFluor^®^ 594 probe (1:100 in BSA 1% in PBS). As primary antibodies, we used mouse anti-DRP1 (1:150), rabbit anti-MFN-2 (1:150) (AbCam, Cambridge, UK), mouse anti-PSD-95 (1:200) (Merck Millipore), rabbit anti-Tau (1:1000), rabbit anti-SV-2 (1:1000) (Synaptic Systems) and rabbit anti-synapsin-1 (produced in-house). As secondary antibodies, we used goat anti-rabbit conjugated with AlexaFluor488 (1:1000); goat anti-mouse conjugated with AlexaFluor488 (1:1000); goat anti-mouse conjugated with ALexaFluor594 (1:1000) and goat anti-rabbit conjugated with AlexaFluor405 (1:500) (Thermo Fisher Scientific). All antibodies were diluted in BSA 1% in PBS. Coverslips were washed with PBS followed by 2 min incubation with Hoechst 33,258 dye (1:1000 in BSA 1% in PBS) for nuclear staining. The coverslips were quickly immersed in methanol and mounted in DPX (Sigma-Aldrich). For widefield imaging, we used a Zeiss AxioScope A1 microscope with an AxioCam HRm camera and 40×(air) and 63×(oil) objectives.

### 2.10. Dendrite Extension, Ramification and Spine Density

Dendrite extension and ramification analysis was performed as stated in [37], using the immunofluorescence detection of the cytoskeletal protein MAP-2, known to be located mainly in dendrites and used widely as a neuritic marker. Cells previously plated onto coverslips were fixed for 20 min with freshly prepared 4% paraformaldehyde in PBS and then washed with PBS. This was followed by permeabilization with Triton-X-100 0.2% (Invitrogen) in PBS for 10 min and subsequent blocking with BSA at 3% in PBS for 30 min. Mouse anti-MAP-2 (1:100) was used as the primary antibody, and goat anti-mouse conjugated with AlexaFluor488 (1:1000, Thermo Fisher Scientific) diluted in BSA 1% in PBS was used as the secondary antibody. Coverslips were submerged in PBS for cleaning and then incubated with Hoechst 33,258 dye (1:1000 in BSA 1% in PBS) (Sigma-Aldrich). Then, coverslips were immersed in methanol and mounted in DPX Mountant (Sigma-Aldrich). For dendrite analysis, fluorescence was visualized using the Zeiss AxioScope, as mentioned above. Green-fluorescence images of ten random microscopic fields were acquired per sample. Evaluation of dendrite length and ramification from individual neurons was determined using ImageJ software in the NeuronJ plugin. Spines were assessed by confocal imaging, in a Leica TCS SP8 inverted microscope with a 40×(oil) immersion objective, sequential laser excitation at 405/488/552 nm and spectral detection adjusted for the emissions Alexa Fluor 405/488/594, respectively. Multiple representative images (1080 × 1080-pixel resolution) of at least 3 independent experiments were captured from random microscopic fields. Number of cells, fluorescence area, fluorescence intensity, and number of spines were measured using Fiji software tools [38].

### 2.11. Mitotracker Active Mitochondria Labeling

Cells were incubated for 30 min at 37 °C with 500 nM of MitoTracker Red CMXRos^®^, according to the manufacturer’s instructions (Thermo Fisher Scientific), to stain viable mitochondria, and then fixed with 4% (*w*/*v*) paraformaldehyde [11]. Cell nuclei were stained with Hoechst 33,258 dye. Images were acquired as mentioned previously and total fluorescence intensity (FI) of the Mitotracker Red was assessed using ImageJ software. After all images were scaled, they were converted to black and white (B&W) images (Image > Colour Threshold). FI and cell area were automatically measured using (Analyze > Analyze Particles) with the options “area” and “integrated intensity” selected from the menu “set measurements”. Then, the FI of MitoTracker Red was normalized using the calculated cell size (Appendix A). In total, over 320 cells were processed for each treatment group.

### 2.12. Determination of Aβ1-40 and Aβ1-42 by ELISA

Equal volumes of cell supernatants were collected and immediately stored at −80 °C until analysis. Aβ1-40 and Aβ1-42 release was quantified by sandwich ELISA assay, according to the manufacturer’s instructions (IBL, Fujioka-Shi, Japan). Absorbance at 450 nm was measured at room temperature in a GloMax microplate reader (Promega, Madison, WI, USA).

### 2.13. Statistical Analysis

Group comparisons between different cells/conditions were performed by one-way ANOVA with Bonferroni post-hoc multiple comparisons test. Specific pair-wise comparisons were performed by two-tailed Student’s *t*-test assuming equal or unequal variances, as appropriate. Statistical analysis was performed using GraphPad PRISM 8.0.1 software (GraphPad Software Inc., San Diego, CA, USA), and only differences of *p* < 0.05 were considered significant.

## 3. Results

### 3.1. SWE and PSEN AD Neuronal Models Show Upregulated miR-124, miR-125b and miR-21, but Only miR-124 Is Shuttled from Both Mutated Cells into Their Exosomes

We started by confirming the elevated expression of inflammatory-associated miR-124, miR-21, miR-125b, miR-155 and miR-146a in SH-*SWE* cells differentiated with RA and cultured for 24 h (Figure 1A; at least *p* < 0.05), as we have previously shown in [15]. We observed that such inflamma-miRNAs were indeed all significantly upregulated in these cells. Given the advantage of iPSC-derived neurons in better mimicking the AD pathophysiology [39], we next evaluated whether similar changes in those inflamma-miRNAs were present in the iPSC-derived iNEU-*PSEN* cells from the AD patient, when compared with data from the healthy iPSC-derived iNEU-*WT* cells. We found similarities between SH-*SWE* and iNEU-*PSEN* cells for the elevation of miR-124, miR-125b and miR-21 (Figure 1A; at least *p* < 0.05).

Exosomes are implicated in the clearance and dissemination of AD pathogenic proteins [40] and miRNAs, having a critical impact on the function of recipient cells [10]. Most of the miRNAs are passively released into exosomes, thus reflecting their cellular expression [41]. However, exosomal miRNA content may be dictated by cell target requirements, leading to either a selective retention of miRNAs or to the active release of them. In the present study, exosomes were isolated from the cell secretomes and characterized for their morphology by transmission electron microscopy (TEM) (Figure 1B), as well as for usual protein markers by Western blot (Figure 1C). Exosomes showed a round morphology, and we identified the cup shape of surface-desiccated exosomes in high magnification, as well as the typical presence of ALIX, CD63 and Flotilin-1 proteins (Figure 1C). Though exosomes were isolated from the same volume of cell supernatants (each mL from 5 × 10^4^ cells), the concentration of exosomes by TEM was apparently higher in secretomes from the iNEU cells, either mutated or not. Accordingly, we found at least 2-fold enriched total RNA levels (ng/µL) in these cells, as compared with the SH-derived ones (*p* < 0.01) (Appendix A), while total proteins (µg/µL in three pooled samples) were at least 5-fold increased.

Representative histograms by NTA analysis of exosomes from SH-*WT* and SH-*SWE* samples and from a single exploratory of those from iNEU-*WT* and iNEU-*PSEN* cells are depicted in Figure 1D. The data are in line with the other described results indicating that an enhanced number of exosomes are released by the iNEU cell lines, probably derived from the stressful conditions of their generation [40], and eventually aggravated by the presence of the *PSEN* mutation (1.7-fold elevation vs. respective *WT*; *p* < 0.05) (Appendix A). Differences in the biogenesis of exosomes should be explored in the future for a larger number of samples and extended to other iNEU cell lines derived from other mutations. It is worth noting that some variation in exosome size distribution profiles in the fresh collected samples was observed, though the most prevalent size in all samples was near 100 nm, the most usual in exosomes. We cannot, however, disregard that some aggregation processes may justify the appearance of peaks over 100 nm.

Increased levels of miR-124 and miR-155 in exosomes were found independently of the cell of origin (at least, *p* < 0.05), when compared with those from WT cells (Figure 1E). Among all the miRNAs, miR-124 sorted out as the only one elevated in both cells and exosomes in the two AD in vitro models. It is worth noting that miR-155 seems to be selectively and actively shuttled from the iNEU-*PSEN* cells into exosomes, while its sorting from SH-*SWE* cells appears to occur by a passive mechanism. Considering the consistent elevation of miR-124 in both models and in their exosomes, we next explored the effects of modulating miR-124 expression levels in each of the neuronal AD models.

### 3.2. Tansfection with miR-124 Mimic and Inhibitor Is Effective in Both AD Neuronal Models, though iNEU Cells Show a Higher Resistance

miR-124 was described to have neurogenic and neural differentiation potential [30,42] and to be required for the expression of homeostatic synaptic plasticity by miR-124 transfection. Neuroblastoma cells [2], rat-derived neurospheres [43] and dental pulp stem cells [44] were already used for miR-124 transfection. Specifically, the knockdown and overexpression of miR-124 was earlier performed in differentiated SH-*WT* cells [45]. However, it is not clear from the literature whether such a treatment was already carried out in iPSC-derived neurons.

Here, we used the miR-124 mimic and the miR-124 inhibitor in the SH-*WT*/SH-*SWE* and iNEU-*WT*/iNEU-*PSEN* cells to overexpress and downregulate, respectively, the intracellular levels of miR-124 (Figure 2). The controls included mock samples, as well as “scramble” negative controls (NC) for the inhibitor and mimic, respectively, as schematically represented in Figure 2A. Data for all the controls were like the miR-124 expression levels in the non-treated cells depicted in Figure 1A, and between models, thus discarding artifactual effects in the expression levels of miR-124 by the transfection process. Mock controls reproduced the increased expression of miR-124 observed in the naïve SH-*SWE* and iNEU-*PSEN* cells (Figure 2B; at least *p* < 0.05). Successful transfection for either inhibitor or mimic in WT and mutated cells in both human AD models was achieved. Data obtained with the inhibitor revealed that miR-124 downregulation was similarly attained in the four cell types tested (at least 80% decrease, *p* < 0.01). However, levels of the transfection efficiency with miR-124 mimic were dissimilar in each of the models. Improved transfection efficiency was obtained in SH-*WT* and SH-*SWE* cells (29-fold increase and 70-fold increase, respectively, *p* < 0.01), when compared with iNEU-*WT* and iNEU-*PSEN* cells, suggesting a higher resistance of these cells to the treatment [46]. Here, we reached an overexpression of ~17-fold in iNEU-*WT* and of ~12-fold in iNEU-*PSEN* cells. Therefore, we cannot rule out that such differences in the overexpression levels for miR-124 modulation in each of the two tested AD models may have had an impact on data obtained in cells and miRNA dynamics with miR-124 inhibitor and miR-124 mimic, which will be presented below.

### 3.3. miR-124 Is an Inducer of Neurite Outgrowth and Mitochondria Activation in SH-SWE Cells

miR-124 has been indicated as a promising therapy in neurodegenerative disorders [47] and to attenuate some AD pathological processes [48,49]. Its overexpression in mouse P19 cells promoted neurite outgrowth, while the opposite was observed by blocking miR-124 function [50]. Therefore, we first assessed the impact of miR-124 inhibitor and miR-124 mimic on total dendrite length by using the MAP-2 immunostaining analysis (Figure 3A). While no differences were observed in either iNEU-*WT* or iNEU-*PSEN* cells (data not shown), a decrease in dendrite length was achieved by the transfection with the miR-124 inhibitor in both SH-*WT* and SH-*SWE* cells (Figure 3B; *p* < 0.05). No changes were found for the total dendrite length in SH-*WT* and SH-*SWE* cells between mock controls. In contrast, an increase in total dendrite length was noticed by the modulation with the miR-124 mimic, though only statistically significant in the mutated SH-*SWE* cells (*p* < 0.05). Differences between non-modulated SH-*WT* and SH-*SWE* cells were only observed when the primary dendrite length and the number of ramifications per cell were considered, with a reduction, and an elevation, respectively, only in mutated cells, suggesting an associated increased stress-related condition. The profile of primary dendrite length per cell upon treatment with the miR-124 modulators was like that of the total dendrite assessment (decreased by the inhibitor, *p* < 0.01 in SH-*WT* and *p* < 0.05 in SH-*SWE*; elevated by the mimic only in SH-*SWE*, *p* < 0.01). The branching of neurites increased with the inhibitor (*p* < 0.05) and decreased with the mimic (*p* < 0.05) in SH-*SWE*, but not in SH-*WT* cells.

Efficient mitochondrial function is necessary for neurite outgrowth [51]. To analyze mitochondrial potential and location, we used MitoTracker-Red, a dye that selectively accumulates in metabolically active mitochondria. At first glance, no significant differences were found between the raw mean intensity per cell in each of the modulations performed in SH-*WT* or SH-*SWE* cells (data not shown). However, when normalized to cell area (data are provided in Appendix A), MitoTracker intensity was significantly decreased in SH-*SWE* cells with the inhibitor and enhanced in both cell types with the mimic (Figure 3C; *p* < 0.05), which led to visible MitoTracker staining dots in neuronal projections (white arrowheads, Figure 3A). It is worth noting that, though no significant differences were obtained, we found changes in cell area between the SH-WT and SH-SWE treated with the mock, the inhibitor, and the mimic. However, in the case of SH-WT cells, the increase in the Mitotracker intensity seems to not derive from the cell area, but to be induced by the mimic, considering that the cell area in the mock-treated SH-WT cells (1037 µm^2^) are close to that of the same cells treated with the mimic (961 µm^2^) (Appendix A). Therefore, though we cannot disregard the contribution of cell area differences to Mitotracker Red intensity increase by the mimic, namely, in SH-SWE cells, it cannot be considered the only responsible factor for the findings achieved.

Once mitochondria fusion and fission dynamics influence neuronal processes and cell survival [52], we next assessed MFN-2 and DRP-1, two proteins implicated in such processes [53,54]. The inhibitor of miR-124 did not cause any modification in mitochondria fission dynamics (Figure 3C), whereas it caused an elevation of mitochondria fusion in SH-*SWE* cells, probably to mitigate cell stress induced by the treatment (Figure 3A,C; *p* < 0.05 vs. respective mock and WT cells). Additionally, when cells were assessed for retrograde (dynein) and anterograde (kinesin family number 3A, KIF3A) axonal transport proteins, miR-124 mimic enhanced the expression of both genes in SH-*WT* cells (at least *p* < 0.05) and of KIF3A in iNEU-*WT* ones (data not shown). No changes in such processes were noticed in either the mutated cells or in those treated with the miR-124 inhibitor. It is worth adding that none of the miR-124 modulatory treatments altered cell viability (Appendix A).

### 3.4. miR-124 Expression Levels below a Certain Threshold Compromise Dendritic Spine Density in iPSC-Derived iNeurons

Dendritic spine dysfunction is a critical feature in the pathogenesis of AD-associated dementia [55]. Dendritic spines are specialized structures on neuronal processes whose loss directly correlates with synaptic function failure [56]. Therefore, a better understanding of the dendritic spine alterations may drive the success of therapeutic interventions in AD. Dendritic morphogenesis and spine density were shown to be promoted by miR-124 in mouse olfactory bulb neurons [57]. Here, we profit from having our iNEU-*WT* or iNEU-*PSEN* cell models to explore the consequences of modulating miR-124 expression levels on the number of dendritic spines that receive input from a synapse of another neuron, since such an assessment cannot be realized in SH-SY5Y neuroblastoma cultures [58].

iNEU-maturated cells, presenting clear MAP-2 and Tau immunostainings, as well as the synaptic proteins SV-2, synapsin and PSD-95 (Appendix A), showed a much higher morphological complexity than that of SH-*WT* and SH-*SWE* cells. When cells were assessed for the spine density (Figure 4), no significant differences were observed with the miR-124 mimic, relative to data on the mock control, though a slight elevation was noticed. In contrast, the miR-124 inhibitor led to a significant reduction in the number of dendritic spines, either in iNEU-*WT* (*p* < 0.01) or in iNEU-*PSEN* (*p* < 0.05) cells, suggesting that it is important to sustain miR-124 expression levels within defined limits to hold functional plasticity.

### 3.5. miR-124 Reduces APP Gene Expression, While It Differently Influences Aβ Oligomerization and Tau Phosphorylation in Each of the Experimental AD Neuronal Models

The data obtained so far suggest that decreased levels of miR-124 induces a decrease in neurite length and number of spines. BACE1 and Tau were found to be decreased in the APP/PS1 mice after treatment with miR-124 [59], and biopsy samples from sporadic AD patients revealed reduced levels of miR-124 and increased gene and protein expression of BACE1 [8]. However, miR-124 was also found to be increased in the brains of AD patients and in male Tg2576 transgenic mice [9]. Such discrepancies accounted for our interest in investigating how the different expression levels of miR-124 could modify APP processing and Tau hyperphosphorylation in our AD cell models. We started by characterizing such AD hallmarks in the non-modulated cells. We observed an upregulation of the APP gene expression in both mutated cells relatively to their matched WT controls (Figure 5A; *p* < 0.01). miR-124 inhibitor induced an increase in the APP mRNA expression in the SH-*WT* cells (*p* < 0.05), but not in any of the other cells. Most interestingly, the miR-124 mimic markedly decreased APP mRNA levels in both SH-*SWE* (*p* < 0.05) and iNEU-*PSEN* (*p* < 0.01) cells.

Aβ oligomers are known to be increased in the early stages of AD and to attack neurons [60], while also inducing Tau hyperphosphorylation [61], namely, at serine residues 396 and 404 [62], that contribute to microtubule and synapse dysfunctions [63]. The SH-*SWE* cells revealed increased levels of the real toxic Aβ trimers/tetramers species (~20 kDa), relatively to SH-*WT* cells, which were sustained by treatment with the miR-124 inhibitor (Figure 5B,C; *p* < 0.05). Notoriously, miR-124 mimic was able to significantly reduce small Aβ oligomer formation, when compared with the mock control (*p* < 0.05). Oligomerization was, however, different in iNEU-*PSEN* cells, where we observed a predominance of larger oligomers (less neuroactive but can dissociate into small oligomers) that persisted with each of the tested modulations (*p* < 0.05). However, no differences for the release of Aβ1-40 or Aβ1-42 from the mutated cells were observed, either before or after miR-124 modulation, except for the enrichment of Aβ1-40 in the secretome from SH-*SWE* cells, as compared with that from the SH-*WT* ones (Appendix A; *p* < 0.05).

Next, Tau phosphorylation at the serine 404 site was evaluated. We could not detect any basal difference in the phosphorylated Tau between WT and mutated samples of each model, though the ratio between the phosphorylated Tau and the total one was near 1-fold in the neuroblastoma cells and 2-fold in the iPSC-derived neurons. Inhibition of miR-124 was translated into an increase in such a ratio only in SH-*WT* cells vs. mock control (Figure 5D,E; *p* < 0.05). Curiously, when we upregulated miR-124 with its mimic, the ratio decreased to 1 only in iNEU-*PSEN* cells (*p* < 0.01), pointing to miR-124 benefits in decreasing the amount of the phosphorylated Tau. Altogether, the data suggest that the upregulation of miR-124 prevents some of the AD-associated risk factors.

### 3.6. Neuron-to-Neuron Transfer of miR-124 Is More Effective among SH-SWE Cells Than between SH-WT Cells

Considering the benefits of the upregulated miR-124 in reducing APP processing, we wondered whether SH-*WT* and SH-*SWE* cells treated with miR-124 mimic would transmit this miRNA to neighboring neurons. If so, that may have potential benefits on using such a strategy to counteract AD pathological mechanisms. The shuttle of miRNAs from cell-to-cell was already shown and suggested to be mediated by exosomes and vesicle-free extracellular miRNAs [64]. In this regard, we decided to explore this issue using SH-*WT* and SH-*SWE* cells in a non-cell-contact co-culture system (Figure 6). For that, mock and miR-124 mimic transfected SH-*WT* as donor cells were co-cultured with matched non-treated cells for 24 h. The same was performed with the SH-*SWE* cells. We noticed that miR-124 expression in transfected SH-*WT* and SH-*SWE* donor cells decreased with the time of coculturing (more than 60% after 24 h), by comparing the transfection efficiency in Figure 6 with that previously obtained in monocultures (Figure 2). Despite the potential transfer of the miR-124 into the secretome (as a soluble species or as part of the exosomal cargo), and to the non-treated cells, we still observed increased levels of miR-124 in the original donor cells relatively to mock controls (14-fold, *p* < 0.01) after the 24 h of incubation. As expected, the co-culture of mock-treated SH-*WT* and SH-*SWE* cells with the respective non-treated cells did not cause any change in their miR-124 expression.

Interestingly, a 5-fold upregulation of miR-124 in SH-*WT* naïve recipient cells was obtained after being co-cultured with the miR-124 mimic transfected SH-*WT* donor cells (*p* < 0.05 vs. mock transfected donor cells co-cultured with SH-*WT* naïve recipient cells), though the transference corresponded to only 40% of the directed transfected cells (*p* < 0.01 vs. miR-124-transfected donor cells). The same was observed in SH-*SWE* cells, with a 3-fold increase in the recipient cells, i.e., near 60% of that in donor cells transfected with miR-124 mimic (*p* < 0.01 vs. mock transfected donor cells co-cultured with SH-*SWE* non-transfected recipient cells).

From these data, we believe that miR-124 is indeed transmitted from neuron to neuron, though apparently with more efficiency in SH-*SWE* than in SH-*WT* cells, perhaps by easy cell-to-cell propagation of miR-124 between mutated cells. Only future studies will confirm if such a propensity is also true for other miRNAs.

### 3.7. Extracellular miRNA Signaling after Neuronal miR-124 Modulation with Its Mimic or Inhibitor Depends on the Cell Pathological Signature and Associated Mutation

After confirming that miR-124 may disseminate from neuron to neuron, we explored the contribution mediated by exosomes and by the vesicle-free secretome in such propagation. In fact, miRNAs can be present in non-exosomal free-floating species and complexes, including in free Argonaut 2 complexes that account for more than 90% of all extracellular miRNAs, or in exosomes [65,66]. As depicted in Figure 7A, the distribution of the upregulated miR-124 in SH-*WT*, SH-*SWE*, iNEU-*WT* and iNEU-*PSEN* cells occurs by soluble non-exosomal (at least *p* < 0.05, except for SH-*WT* cells) and exosomal dissemination (*p* < 0.01 for all). It is also worth noting that miR-124 inhibition was similarly reflected by its reduced levels in soluble and in exosomal fractions from the transfected SH-*SWE* cells (*p* < 0.05) and in the iNEU-*WT* soluble fraction (*p* < 0.05). The results confirm that we may enrich or deplete the extracellular secretome cargo in miR-124 by up- or downregulating its expression levels in donor cells. Efficiency was more notorious for the upregulation of miR-124, namely, in the exosomal component, attesting its passive delivery from the transfected donor cells.

Then, we investigated whether the transfection with miR-124 mimic and miR-124 inhibitor in SH-*WT*/SH-*SWE* and in iNEU-*WT*/iNEU-*PSEN* cells would have an influence on cellular and exosomal inflamma-miRNA signature (Figure 7B–D for the mutated cell responses to the modulation; Appendix A for the WT cell responses to the modulation).

Decreased miR-125b levels were generally observed in the mutated cells and in their exosomes with both inhibitor and mimic (at least *p* < 0.05) (Figure 7B). Different responses were obtained in WT cells and respective exosomes, since low levels were observed with the miR-124 inhibitor (at least *p* < 0.05, except for SH-SWE exosomes), but increased ones with the miR-124 mimic (*p* < 0.01, except for iNEU-WT cells), as depicted in Appendix A.

Relatively to miR-21, it behaved diversely in each of the AD cellular models upon the modulation with the inhibitor. It increased in SH-*SWE* cells (*p* < 0.05) and was depleted in exosomes (*p* < 0.01), pointing to an active retention by the cells, while it was decreased in iNEU-*PSEN* cells (*p* < 0.01), without changes in the exosomal cargo. The miR-124 mimic caused a depletion of miR-21 in both AD models, but only significant in iNEU-*PSEN* cells (*p* < 0.01) and in their exosomes (*p* < 0.05). The representation of miR-21 in WT cells after modulation with the miR-124 inhibitor (Appendix A) was the opposite of data in the mutated cells, with no changes in the SH-*WT* cells, but a significant increase in the iNEU-*WT* cells (*p* < 0.01), and similarly decreased exosomal levels. Inverse data were obtained with the miR-124 mimic. It caused miR-21 upregulation in SH-*WT* cells and in exosomes from both cell lines (at least *p* < 0.05).

miR-146a was found to be upregulated in SH-*SWE* and iNEU-*PSEN* cells upon the modulation with the miR-124 inhibitor (*p* < 0.01) (Figure 7B). However, it was only enriched in exosomes from SH-*SWE* cells, once it was found to be depleted in those from iNEU-*PSEN* cells (*p* < 0.01) (Figure 7C). The retention of miR-146a in these mutated cells may derive from their specific requirements. In WT cells, the elevation of miR-146a by the miR-124 inhibitor was only observed in SH-*WT* cells (*p* < 0.05), and its presence in exosomes was mostly reduced (Appendix A). A general decline upon the miR-124 mimic was observed in the mutated cells and in their exosomes (at least *p* < 0.05, except for iNEU-*PSEN* cells), as well as in the WT cells (at least *p* < 0.05), but then with an increased exosomal representation (at least *p* < 0.05). These data suggest that cells manipulate the passive and the selective release of miRNAs differently, in a target-dependent manner and accordingly to specific pathological features.

Finally, miR-155 was unchanged by the miR-124 inhibitor in the mutated cells/exosomes. However, we found its elevation in the WT cells and their exosomes (at least *p* < 0.05, Appendix A), except in iNEU-*WT* cells, where a reduction was obtained (*p* < 0.01). In turn, the miR-124 mimic led to a generalized reduction in miR-155 in the mutated and *WT* cells/exosomes (*p* < 0.01, except for iNEU-*PSEN* and SH-*WT* cells), thus indicating a preventive effect of the miR-124 mimic in miR-155-mediated neuroinflammation.

Other contrasting differences between the expression of inflamma-miRNA profiles after mimic and inhibitor modulations in cells and exosomes were obtained when the two-tailed Student’s *t*-test was used and are portrayed in Appendix A.

Given the pleiotropic differences in mRNA signaling obtained between *WT* and mutated cells, with either inhibitor or mimic, together with their potential dependence on models and cell mutations explored in the present study, we decided to summarize the results in Figure 7D for clarity. Our results emphasize the highly dynamic network of miRNA gene regulation and its dependence on the cell requirements to sustain homeostasis, which varied with the genetic mutation and the experimental neuronal cell lines. Data also support the benefits of miR-124 modulation in sustaining neuronal dynamics and balanced miRNA signaling in AD and the use of iNeurons in modelling AD disease, though it is worth highlighting that just a cell line was investigated, thus not representing the heterogeneity of AD patients.

## 4. Discussion

Experimental models of AD are crucial to gaining better knowledge on the pathological processes and to test the potential of new therapeutic approaches. However, in the case of transgenic mice, the high failure rate in the translation of successful results into clinics outlines their key limitations [31]. The use of human cell-based experimental models surpasses the concerns associated to such species differences. Neuroblastoma cells are one of the most widely used in vitro model of AD, though not fully reproducing Aβ and Tau pathology [67]. Some of these flaws have been tentatively surpassed by using the SH-SY5Y cell line overexpressing either WT human APP (*APP695*) or *APP harboring the Swedish double mutation* (*K670N/M671L*) [68]. In conformity, here, we used such human AD neuronal cells since they are cost-effective, easy to work, can be kept in culture for long periods of time, are more precise, provide more material and bypass ethical concerns. Interestingly, we have previously shown that these cells release Aβ1-40 [15] and may be important to accelerating therapeutic drug discovery and saving money, before moving to animal models or other more advanced systems to validate findings. Alternatively, iPSCs started to be generated from patient fibroblasts, and the use of iPSCs-differentiated neurons allowed for studies on sporadic and familiar cases of AD that better mimic the pathological condition [69]. However, they also show some limitations, which are mostly associated to a difficult recapitulation of cell ageing, phenotypic variation, and an absence of interactions with glial cells, if not used in triculture cell types and 3D models. We can expect that these constraints will be overcome in future studies using the direct conversion of somatic cells [70].

The modulation of miR-124 in the two different AD neuronal models has benefits on the prevention of AD-associated pathological mechanisms, which may, however, vary across the in vitro models and the human iPSC-derived neuronal lines from either sporadic patients or presenting diverse mutations. In conformity, our study has limitations by just using a control case and one iNEU-*PSEN* cell line, which may represent only part of the AD spectrum. Nevertheless, our results underline some improvements by the miR-124 upregulation and indicate that the most important will be to keep miR-124 levels between strict limits to sustain cell homeostasis and to adopt a modulatory strategy that considers the patient miR-124 signature, thus contributing to a breakthrough in personalized medicine.

The *PSEN1* mutant iPSCs, from which we derived our iNEU-*PSEN*, manifested hallmarks of AD pathology (increased Aβ production, oxidative stress and compromised neurosupportive function) when differentiated into astrocytes [32]. Interestingly, when differentiated into microglia, despite evidencing enhanced migration ability, the cells released diminished levels of cytokines [33]. These authors evidenced heterogeneous microglia phenotypes depending on whether they derived from iPSCs associated with the *APOE4* genotype, or with the *APPSwe* and *PSEN1* mutations, supporting that those microglia impairments are somehow associated to the implicated mutations. Thus, we believe that more pertinent findings can be obtained when combined experimental models are used to address controversial issues, as is the case with miR-124 in AD pathogenicity. Actually, miR-124 and its role in neuropathology is a hot topic with accumulating contradicting reports of increased [9] and decreased [8] levels in AD, as well as different perspectives about the protective or detrimental effects.

We hypothesize that such miR-124 inconsistencies derive from studies performed with different experimental models, disease stages and brain regions [10]. In sum, these studies have not, so far, provided a clear idea on the role of miR-124 in the AD onset and progression, nor if discrepancies that were found derive from functional neuronal differences among sporadic and familial AD patients, supporting the urgent need for reliable biomarkers for predisposing susceptibilities and pathology staging.

The present study then used the SH-*SWE* cells, already explored by us in previous studies [15] and the new iNEU-*PSEN* hippocampal neurons derived from iPSCs of an AD patient based on stablished protocols [32,35]. In such prior work, we have found that miR-124, miR-21 and miR-125b were upregulated in SH-*SWE* cells and their exosomes, and that increased neuronal miR-124 contributed to its increased expression levels in microglial cells, after 48 and 72 h of coculturing, leading to a decrease in the miR-124 target CCAAT/Enhancer-Binding Protein alpha (C/EBP-α) mRNA expression. However, as far as we know, there are no data concerning the expression of such miRNAs in neurons generated from iPSCs, mainly if considering the *PSEN1* mutation. Moreover, the existing studies on the miRNA role, namely, that of miR-124, in AD pathogenicity are not yet consensual, as mentioned above.

Therefore, we assessed our set of inflammatory-associated miRNAs in SH-*SWE* cells and their exosomes and compared with data from the same evaluation in iNEU-*PSEN* cells, as well as from their respective controls SH-*WT* and iNEU-*WT* cells. We found the presence of miR-124, miR-125b and miR-21 upregulations in both SH-*SWE* and iNEU-*PSEN* cells, while in exosomes there was an increased representation of miR-124 and miR-155 in the two mutated cell types. However, only miR-124 was consistently enriched in cells and exosomes in both models, indicating a crucial role in neuronal function/dysfunction and paracrine signaling. Our data are in line with other findings obtained in AD patients and in the *Tg2576* transgenic mice, where miR-124 was also found to be overexpressed [9,10]. Altogether, such findings enhanced our interest to explore the beneficial and detrimental aspects of miR-124 expression levels in AD pathology in these two cell experimental models, the SH-*SWE* and iNEU-*PSEN* cells and in their derived exosomes. For that, we decided to overexpress and downregulate miR-124 by using its mimic and inhibitor. In this context, our study was pioneering; miR-124 modulation in neurons had heretofore only been tested once in SH-SY5Y cells [45]. The transfection of cells resulted in the upregulation of miR-124 by the mimic and downregulation by its inhibitor. Slightly higher resistance to transfection by iNEU-*PSEN* cells relatively to SH-*SWE* may relate with a more difficult process in primary and stem cells when compared with the “easier” efficiency achieved in cell lines [71,72].

One of the known effects of miR-124 expression in neurons is to promote neurite outgrowth by regulating genes encoding proteins associated to cytoskeleton organization [50]. Such a finding was observed by the miR-124 mimic, mainly in the SW-*SWE* cells, but such influence was not noticed in the INEU-*PSEN* cells. Curiously, miR-124, together with miR-9, were shown to also increase dendritic complexity in cortical neurons obtained from embryonic mice [73]. We did not observe any modification in the SH-*WT* cells, while increased ramification was noticed in the SW-*SWE* cells, either untreated or treated with the miR-124 inhibitor, returning to control levels with the miR-124 mimic. This is in accordance with the role of miR-124 in suppressing variability in dendritic branching numbers [74]. Efficient mitochondrial function is the basis for successful neurite outgrowth [51]. Increased red fluorescence intensity of MitoTracker staining was observed following miR-124 mimic transfection in either SH-*WT* or SH-*SWE* cells, probably to assure the enhanced length of dendrites, a process that demands high energy [75]. It is worth noting, however, that apart from the inhibitor-dependent enhanced fusion in SH-*SWE* cells, no significant changes were noticed for DRP1, thus suggesting the absence of mitochondria structural abnormalities or elevated cytosolic Ca^2+^ levels [76].

Synaptic failure is a cause of cognitive decline, and the loss of dendritic spines is associated to synaptic dysfunction [56] and the pathological accumulation of Tau in AD [77]. In the present study, the loss of dendritic spines was evidenced by the miR-124 inhibitor only in iNEU-*WT* and iNEU-*PSEN* cells once they could not be visualized in neuroblastoma cultures [58]. Increased Tau phosphorylation was observed by the miR-124-inhibitor in SH-*WT* cells, but its decrease was achieved by the mimic in iNEU-*PSEN* cells, suggesting a potential beneficial effect in AD pathology, already proposed in a previous study [49]. In addition, data also show that iNEU-*PSEN* cells are a reasonable alternative AD model to the transgenic mice to test strategies for attenuating Tau hyperphosphorylation and support the relevance of sustaining miR-124 expression levels within defined margins, as already proposed by other Authors [78]. We may also emphasize that the neuroblastoma model is more satisfactory to evaluate impacts on dendrite outgrowth, while iPSC-derived neurons are a better alternative for assessing dendritic spine impairments (Table 1).

APP upregulation is known to promote AD pathogenesis by favoring Aβ generation [79], which is still considered a primary pathological hallmark of AD [80]. Elevated expression levels of APP was a common finding in the two AD human cell models here used. At least in SH-*SWE* cells, we demonstrated that the reduction in miR-124 expression was translated into an increased *APP* mRNA level, thus promoting Aβ production and emphasizing its detrimental outcome. It is worth highlighting, however, that the upregulated expression of miR-124 when we used its mimic was particularly beneficial in counteracting *APP* gene expression in the mutated cells, either in SH-*SWE* or in iNEU-*PSEN* ones. This feature, again, validates the need for regulating miR-124 levels between narrow thresholds when one considers the fact that BACE1 is a potential downstream target of miR-124 [59]. Light Aβ species formed by trimers/tetramers were noticed in the untreated SH-*SWE* cells, while the oligomeric Aβ species were predominant in the iNEU-*PSEN* cells (Table 1). Both Aβ trimers/tetramers and oligomers are considered neurotoxic species [81,82] and, though distinctly represented, were found in the differently mutated cells.

In the past years, it was demonstrated that miRNAs are not only present intracellularly, but are also detectable in substantial amounts extracellularly, including in biological fluids, being present either in exosomes or as soluble components in the vesicle-free secretome [83]. This horizontal transfer of miRNAs is part of autocrine and paracrine signaling mechanisms involved in cell-to-cell communication and in the propagation of inflammatory mediators, including miRNAs [11,84,85]. In this study, we have shown that SH-*WT* and SH-*SWE* cells overexpressing miR-124 are responsible for its enhanced expression in donor cells, when using a non-contact co-culture system, thus attesting the cell-to-cell transfer of this miRNA [64]. To further explore this issue, we assessed the presence of miR-124 in either the soluble part of the secretome or as part of the exosomal cargo when released from SH-*WT*/SH-*SWE* and iNEU-*WT*/iNEU-*PSEN* cells treated with the miR-124 mimic. As expected, we identified miR-124 presence as a soluble and exosomal component, when derived either from *WT* cells or from mutated ones. However, miR-124 was highly enriched in exosomes from iNEU-*PSEN* cells comparatively to those from SH-*SWE* cells, thus indicating its preferential exosomal package in such case. Though the favored loading of specific miRNAs into exosomes was noticed for other miRNAs, such as miR-320 and miR-150, the decision of the host cells for active or passive miRNA sorting processes into exosomes remain a controversial issue [86]. When considering the effects exerted by the intracellular miR-124 inhibitor in the respective exosomal cargo, one major difference between the SH-*SWE* and the iNEU-*PSEN* cells was found for miR-146a expression levels. While increased in both cell lines, its sorting was processed by a passive way in the first case, while it was actively retained in iNEU-*PSEN* cells, probably due to its increased target genes [41]. No major differences were observed between the action of the miR-124 inhibitor or the miR-124 mimic on the expression of miR-125b, which was decreased in both cases. Considering that miR-125b induces Tau hyperphosphorylation [87], one may consider that both modulatory interventions may have benefits. Another interesting finding was that miR-124 mimic led to a decrease in miR-155 and miR-146a in exosomes released by either SH-*SWE* or iNEU-*PSEN* cells, contributing to a less inflammatory milieu considering their role as pro-inflammatory mediators [10,88]. Indeed, deletion of miR-155 in mice was shown to promote axon growth, to reduce neuroinflammation, as well as to enhance neuroprotection and repair after injury [89]. Although having a dynamic expression during AD progression, miR-146a inversely correlates with the receptor expression on myeloid cells 2 (TREM2) in patients with AD, which may impair microglia phagocytic ability [90]. Further investigation using co-culture experiments will provide insights into the effects of modulating neurons with miR-124 inhibitor and miR-124 mimic toward microglia polarization into AD disease-associated phenotypes or into reparative subtypes.

## 5. Conclusions

Our study, by using SH-*SWE* (classical) and iNEU-*PSEN* (advanced) cells, was the first to show mutation-independent increased neuronal levels of miR-124, miR-125b and miR-21 and of exosomal miR-124 and miR-155 in both AD models. Among all these intracellular upregulated inflamma-miRNAs, only miR-124 was shuttled into exosomes in the two experimental systems, thus suggesting having an autocrine/paracrine regulation role. Additionally, and though deserving further research, we demonstrate that miR-124 inhibitor largely compromises dendrite length and spines, enhances APP processing, and promotes neuronal miR-146a-induced inflammation. In contrast, such effects are mostly counteracted by the miR-124 mimic, which increased the transmembrane mitochondrial potential, reduced Tau hyperphosphorylation, as well decreased the exosomal cargo in miR-155 and miR-146a. The data suggest that miR-124 expression levels should then be strictly regulated to exert a neuroprotective role. Besides the common findings of miR-124 modulation in SH-*SWE* and iNEU-*PSEN* cells, different responsive profiles with its mimic and/or inhibitor were observed, as highlighted in Table 1, which may relate to the mutation involved or be directly associated to each of the AD neuronal models. These findings may help on the decision to use SH-*SWE* and iNEU-*PSEN* cells depending on the studies intended to be carried out. They also highlight that patient-specific inflamma-miRNA signatures support the application of precision medicine principles in AD. Collectively, our data indicate that miR-124 is a novel diagnostic biomarker in circulating exosomes and a potential therapeutic target in AD, whenever its deregulation is identified. Experiments in the future using neuronal exosomes as vehicles to deliver exogenous miR-124 mimic or miR-124 inhibitor into microglia may contribute to also make these cells attractive for therapeutic modulations in AD.

## Figures and Tables

**Figure 1 cells-10-02424-f001:**
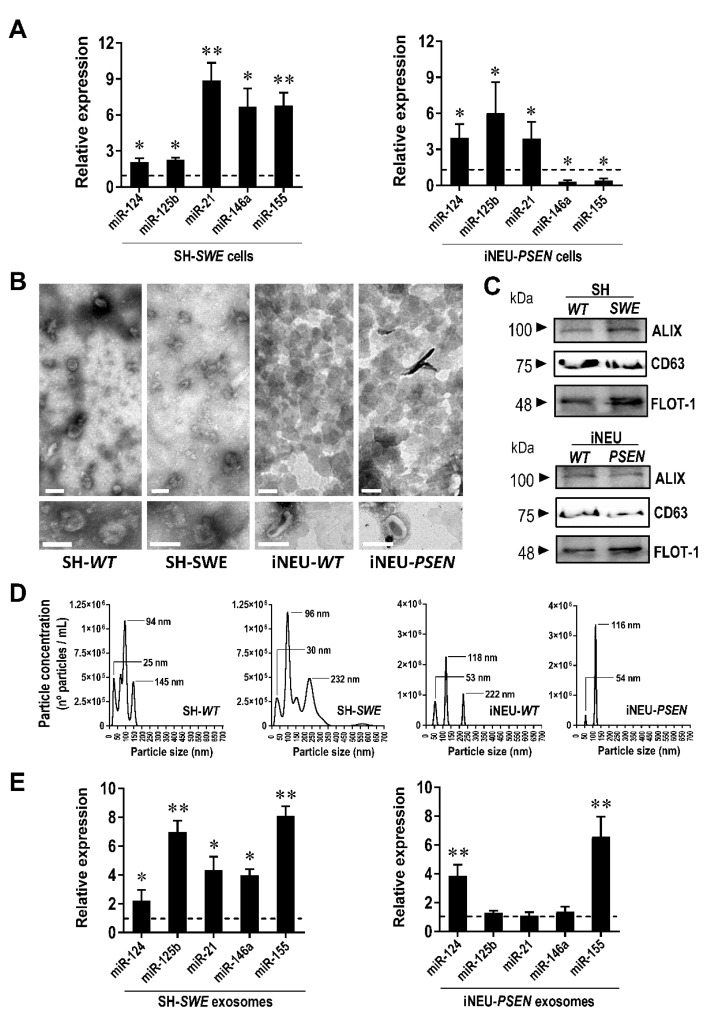
Comparison of inflammatory-associated miRNA (inflamma-miRNA) profiles in SH-*SWE* and iNEU-*PSEN* cells, and in their respective exosomes, relatively to their SH-*WT* and iNEU-*WT* control samples. Cells were obtained and differentiated as described in Material and Methods. (**A**) Inflamma-miRNAs expression levels in both cellular AD models reveal that miR-124, miR-125b and miR-21 are increased. (**B**) Representative Transmission Electron Microscopy (TEM) images of exosomes with characteristic sizes after their isolation from cell secretomes by differential ultracentrifugation. (**C**) Representative Western blots for the common exosomal protein markers, Alix, CD63 and Flotilin-1 (FLOT-1). (**D**) Representative histograms of size and concentration of exosomes per sample. (**E**) Profiles of inflamma-miRNA cargo in exosomes from the different mutated cells evidencing increased expression levels of miR-124 and miR-155. Results are mean ± SEM fold change from at least three independent experiments (except NTA single data for exosomes from iNEU cell lines). * *p* < 0.05 and ** *p* < 0.01 vs. respective WT levels (dashed lines), two-tailed student’s *t* test. miR, miRNA; SH-*WT*, human SH-SY5Y wild-type neurons; SH-*SWE*, human SH-SY5Y expressing the *APP695 Swedish* mutant protein; iNEU-*WT*, iNeurons derived from induced pluripotent stem cells (iPSCs) generated from a healthy control; iNEU-*PSEN*, iNeurons from iPSCs generated from a patient carrying the *PSEN1ΔE9* mutation.

**Figure 2 cells-10-02424-f002:**
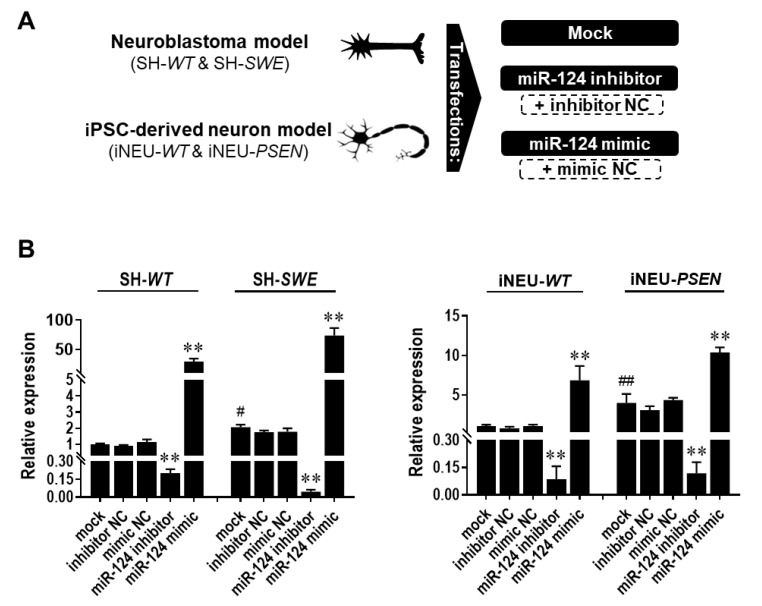
Expression levels of miR-124 in SH-*WT*/SH-*SWE* and iNEU-*WT*/iNEU-*PSEN* cells after transfection with the mock and with the miR-124 inhibitor and mimic. Cells were obtained, differentiated, and transfected, as detailed in Material and Methods. (**A**) Schematic representation of the miR-124 modulation with miR-124 inhibitor and/or mimic, as well as with scramble (respective negative controls—NC). (**B**) miR-124 expression levels before and after its modulation in SH-*WT*/SH-*SWE* and iNEU-*WT*/iNEU-*PSEN* cells. Results are mean ± SEM fold change from at least three independent experiments. ** *p* < 0.01 vs. respective mock control cells, one-way ANOVA with Bonferroni post-hoc test; ^#^
*p* < 0.05 and ^##^
*p* < 0.01, between WT and AD mock controls, two-tailed student’s *t*-test. miR, miRNA; NC, negative control; SH-*WT*, human SH-SY5Y wild-type neurons; SH-*SWE*, human SH-SY5Y expressing the *APP695 Swedish* mutant protein; iNEU-*WT*, iNeurons derived from induced pluripotent stem cells (iPSCs) generated from a healthy control; iNEU-*PSEN*, iNeurons from iPSCs generated from a patient carrying the *PSEN1ΔE9* mutation.

**Figure 3 cells-10-02424-f003:**
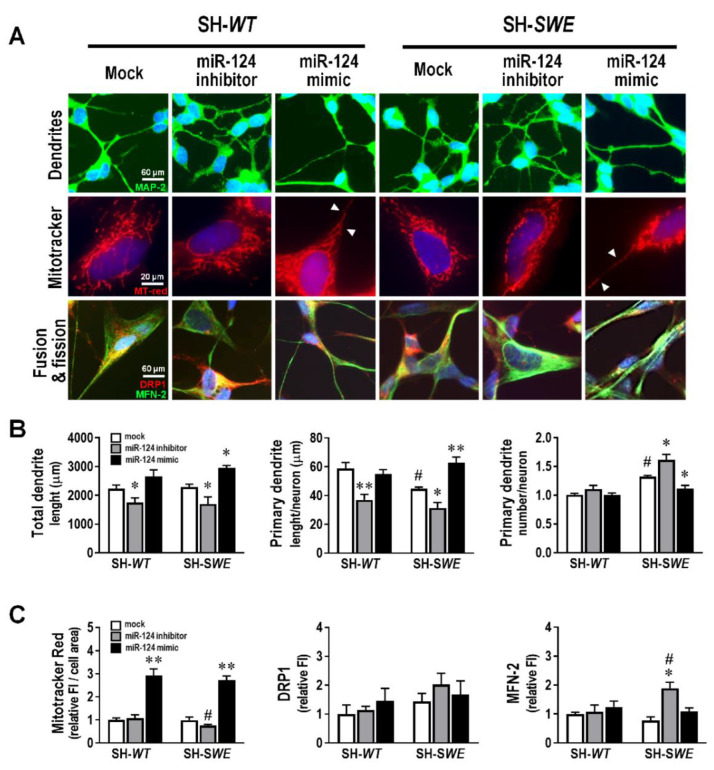
Changes in dendrite outgrowth, mitochondria activation and fusion/fission dynamics by modulation of miR-124 with inhibitor and mimic, relatively to mock, in SH-*WT* and SH-*SWE* cells. Cells were obtained, differentiated, and transfected, as detailed in Material and Methods. (**A**) Representative fluorescence images of MAP-2 (green) for dendrites, Mitotracker (red) for mitochondria, and MFN-2 (green)/DRP-1 (red) for fusion/fission mitochondria dynamics. Cell nuclei are stained with Hoechst 33,258 dye (blue). Mitochondrial clumps are seen in the neurites (white arrowheads). (**B**) Evaluation of total dendrite length, primary dendrite length and primary dendrite number. (**C**) Quantification of fluorescence intensities for Mitotracker Red, DRP-1 and MFN-2. Results are mean ± SEM fold change, from at least three independent experiments. * *p* < 0.05 and ** *p* < 0.01 vs. respective mock controls, one-way ANOVA with Bonferroni post-hoc test. ^#^
*p* < 0.05, between the same treatment in SH-*WT* and SH-*SWE* cells, two-tailed student’s *t*-test. miR, miRNA; SH-*WT*, human SH-SY5Y wild-type neurons; SH-*SWE*, human SH-SY5Y expressing the *APP695 Swedish* mutant protein; iNEU-*WT*, iNeurons derived from induced pluripotent stem cells (iPSCs) generated from a healthy control; iNEU-*PSEN*, iNeurons from iPSCs generated from a patient carrying the *PSEN1Δ E9* mutation; MAP-2, microtubule-associated protein 2; DRP1, protein dynamin-related protein 1; MFN-2, mitofusin-2 protein; FI, fluorescence intensity.

**Figure 4 cells-10-02424-f004:**
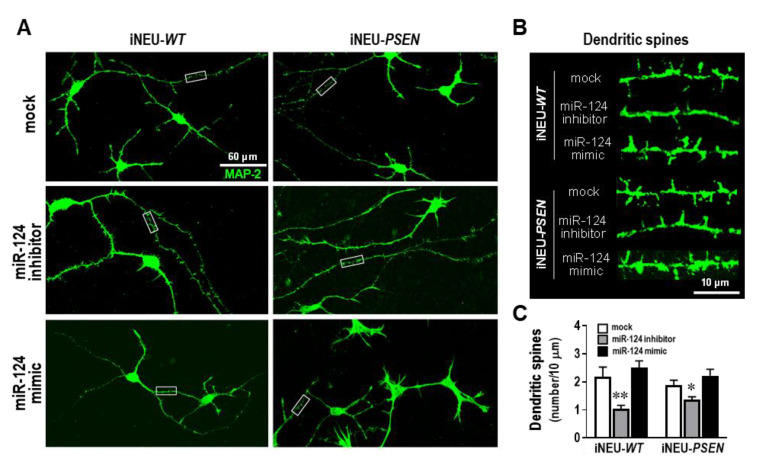
Dendritic spine number decrease in iNEU-*WT* and iNEU-*PSEN* cells by modulation with the miR-124 inhibitor, but no changes were noticed for the mimic relatively to the mock transfection. Cells were obtained, differentiated, and transfected, as detailed in Material and Methods. (**A**) Representative fluorescence images of MAP-2 (green) staining in axons and dendrites in the different conditions. (**B**) Representative images of high magnification of dendritic segments (from panel A insets) with the miR-124 modulation. (**C**) Quantification of the dendritic spine number in each of the conditions. Results are mean ± SEM from at least three independent experiments. * *p* < 0.05 and ** *p* < 0.01 vs. respective mock controls, one-way ANOVA with Bonferroni post-hoc test. miR, miRNA; iNEU-*WT*, iNeurons derived from induced pluripotent stem cells (iPSCs) generated from a healthy control; iNEU-*PSEN*, iNeurons from iPSCs generated from a patient carrying the *PSEN1 ΔE9* mutation.

**Figure 5 cells-10-02424-f005:**
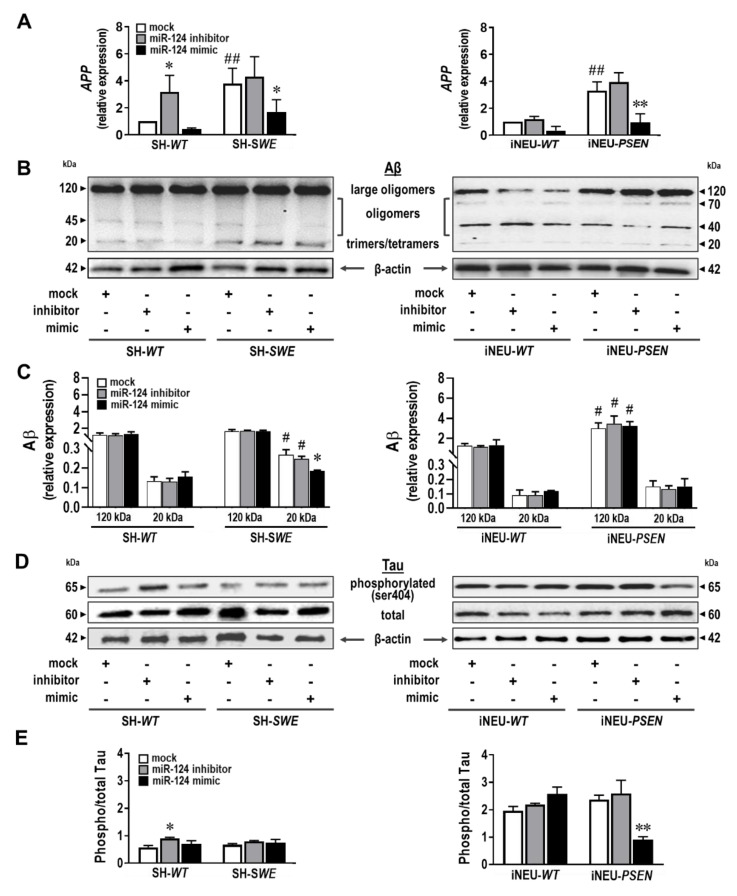
Effects of miR-124 modulation on APP transcription, Aβ accumulation and Tau phosphorylation in SH-*WT*/SH-*SWE* and iNEU-*WT*/iNEU-*PSEN* cells. Cells were obtained, differentiated, and transfected, as detailed in Material and Methods. (**A**) Cellular quantification of APP gene expression by RT-qPCR revealing a decrease in the mutated cells with the miR-124 inhibitor. (**B**) Representative Western blots for Aβ species showing different representation and predominance of bands at 120 and 20 kDa. (**C**) Quantification of small and large Aβ oligomers showing that the first are increasingly represented in SH-*SWE* cells and the later in iNEU-*PSEN* cells. (**D**) Representative Western blots for Tau phosphorylation status at serine 404. (**E**) Quantification of phospho-Tau/total-Tau ratio revealing a decrease in iNEU-PSEN cells by the miR-124 mimic. Results are mean ± SEM fold change from at least three independent experiments. * *p* < 0.05 and ** *p* < 0.01 vs. respective mock controls, one-way ANOVA with Bonferroni post-hoc test. ^#^
*p* < 0.05 and ^##^
*p* < 0.05, between the same treatment in WT and mutated cells, two-tailed student’s *t*-test. miR, miRNA; SH-*WT*, human SH-SY5Y wild-type neurons; SH-*SWE*, human SH-SY5Y expressing the *APP695 Swedish* mutant protein; iNEU-*WT*, iNeurons derived from induced pluripotent stem cells (iPSCs) generated from a healthy control; iNEU-*PSEN*, iNeurons from iPSCs generated from a patient carrying the *PSEN1ΔE9* mutation.

**Figure 6 cells-10-02424-f006:**
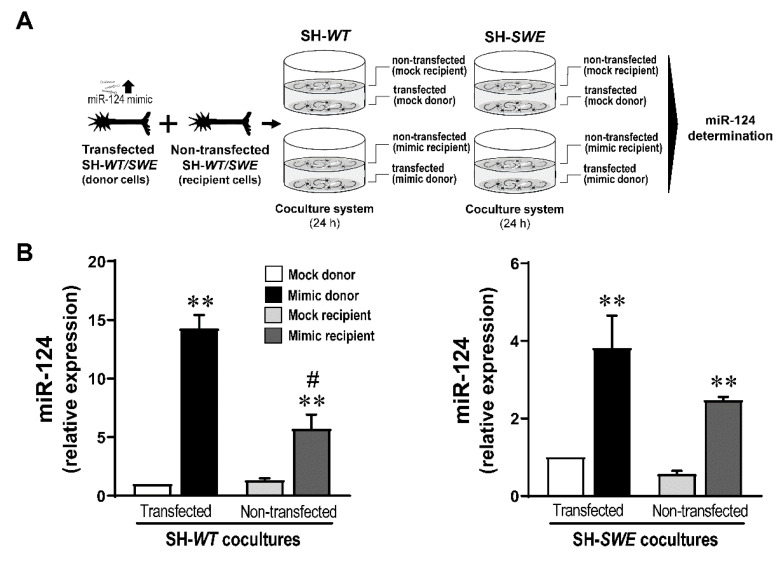
miR-124 is transmitted from donor to recipient neurons in a non-cell-contact co-culture system after transfection of miR-124 in SH-*WT* and SH-*SWE* donor cells. Cells were obtained, differentiated, and transfected, as detailed in Material and Methods, before being co-cultured with non-transfected matched cells (recipient) for 24 h. (**A**) Schematic representation of SH-*WT* and SH-*SWE* donor cells transfected with mock and miR-124 mimic in co-culture with the matched non-transfected recipient cells. (**B**) miR-124 expression levels in the transfected and non-transfected cells after 24 h of co-incubation. Quantification was obtained by RT-qPCR and results are mean ± SEM fold change from at least three independent experiments. ** *p* < 0.01 vs. mock and ^#^
*p* < 0.05 vs. cells modulated with miR-124 mimic, two-tailed student’s *t*-test. miR, miRNA; SH-WT, human SH-SY5Y wild-type neurons; SH-SWE, human SH-SY5Y expressing the *APP695 Swedish* mutant protein.

**Figure 7 cells-10-02424-f007:**
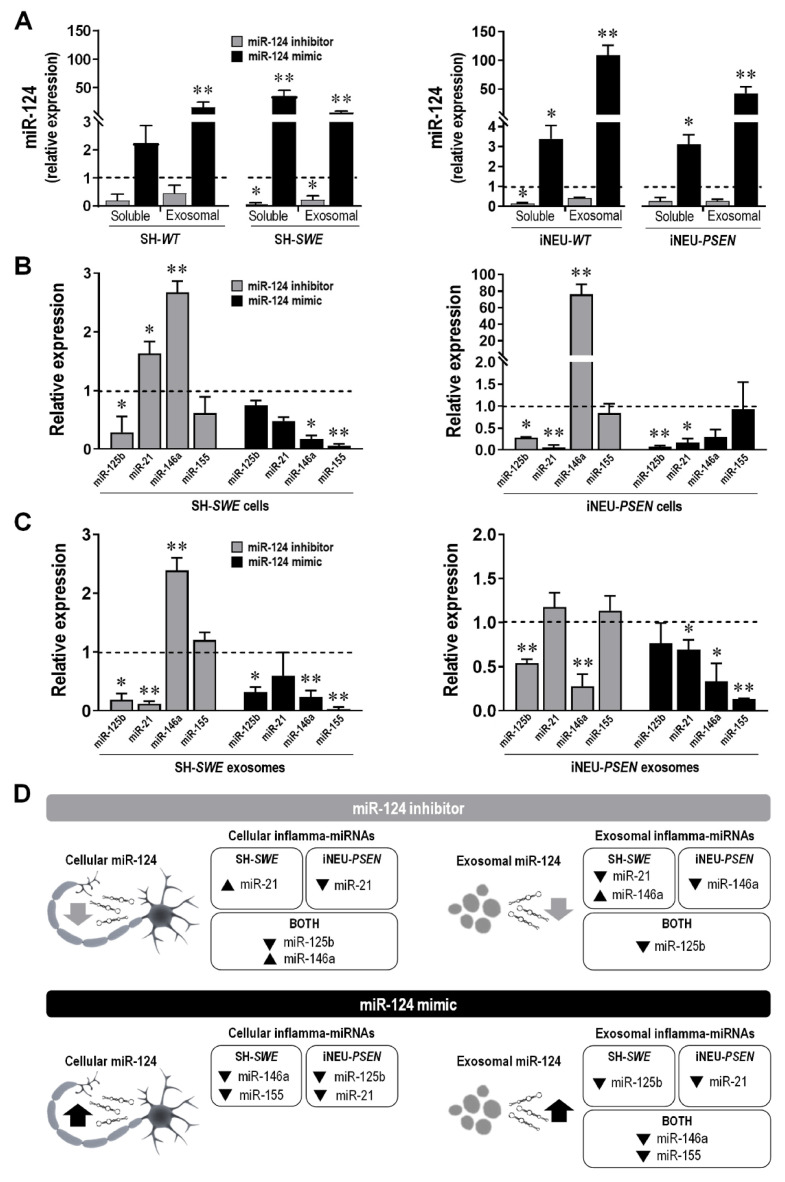
Transfection of SH-*WT*/SH-*SWE* and of iNEU-*WT*/iNEU-*PSEN* cells with miR-124 inhibitor and miR-124 mimic influences miR-124 representation in the vesicle-free secretome (soluble) and packaging into exosomes, while also modifies the representation of other inflamma-miRNAs in the same cells and respective exosomes. (**A**) Evaluation of the influence of the cell treatment with miR-124 inhibitor and miR-124 mimic in the release of miR-124 into the secretome, as a soluble species and as a exosomal cargo. (**B**) Evaluation of cellular inflamma-miRNA expression levels after modulation of miR-124 with its mimic and inhibitor. (**C**) Evaluation of exosomal inflamma-miRNA expression levels after cellular modulation of miR-124 with its mimic and inhibitor. (**D**) Schematic representation of cellular and exosomal inflamma-miRNA distribution in the classical and in the advanced neuronal AD models by miR-124 inhibitor and miR-124 mimic. Results are mean ± SEM fold change from at least three independent experiments. * *p* < 0.05 and ** *p* < 0.01 vs. respective mock controls, one-way ANOVA with Bonferroni post-hoc test. miR, miRNA; SH-*WT*, human SH-SY5Y wild-type neurons; SH-*SWE*, human SH-SY5Y expressing the *APP695 Swedish* mutant protein; iNEU-*WT*, iNeurons derived from induced pluripotent stem cells (iPSCs) generated from a healthy control; iNEU-*PSEN*, iNeurons from iPSCs generated from a patient carrying the *PSEN1ΔE9* mutation.

**Table 1 cells-10-02424-t001:** Comparative pathological characteristics of human SH-*SWE* and iNEU-*PSEN* neuronal AD models in research.

		Pathological Hallmarks in Alzheimer’s Disease Models						
Cell Model	Mutation	miRNAs	Dendrite Length	Spine Dynamics	APP Processing	Intracellular Aβ	Extracellular Aβ	Tau Phosphorylation
SH-*SWE*	*APP KM670/671NL*	Yes	Yes	No	Yes	Yes (small oligomers)	Yes (Aβ1-40)	No
iNEU-*PSEN*	*PSEN1ΔE9*	Yes	No	Yes	Yes	Yes (large oligomers)	No	Yes

SH-*SWE*, human SH-SY5Y expressing the *APP695 Swedish* mutant protein; iNEU-*PSEN*, neurons differentiated from induced pluripotent stem cells generated from a patient carrying the *PSEN1ΔE9* mutation; APP, amyloid precursor protein; Aβ, amyloid-beta peptide.

## Data Availability

The data presented in this study are available on request from the corresponding author.

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
