# Peer review of "Neuronal Dynamics and miRNA Signaling Differ between SH-SY5Y APPSwe and PSEN1 Mutant iPSC-Derived AD Models upon Modulation with miR-124 Mimic and Inhibitor"

_cells, 2021, doi:10.3390/cells10092424_

Round 1

Reviewer 1 Report

In this article, Garcia and colleagues provide the benefits of keeping neuronal miR-124 within strict expression 34 levels, the dependence of miR-124 properties on experimental research models and mutations, and 35 the advantage of iNEU-PSEN cell for AD mechanistic studies and development of personalized 36 therapeutic strategies.

As expected from this group this manuscript represents a strong set of experiments with interesting work. This manuscript is well written, and the data and their statistical analyses were solid. The conclusions made were well supported by several independent experimental approaches. Lastly, the IF images are very beautiful.  

Author Response

R1 – We are grateful for the positive appreciation of our study by the Reviewer. Authors revised the English language and improved the writing of the manuscript.

Reviewer 2 Report

In the present manuscript, the role of miR-124 in the pathogenesis of Alzheimer's disease was studied using two SH-SWE the iNEU-PSEN cell models. The authors showed that miR-124 modulates a number of cellular functions associated with neurite outgrowth and the secretion of inflammatory-associated miRNAs into extracellular vesicles, as well as reduced APP gene expression in neurons. This work has convincingly demonstrated the role of miR-124 in intercellular communication through extracellular vesicles and the need for further studies of the neuroprotective properties of miR-124. The reviewer has several questions requiring revision of the article.

Major comments

The authors make quantitative comparisons of total RNA levels in exosomes obtained from the two types of cell models and found statistically significant differences. However, the obtained data could be explained by the different amounts of secretion of extracellular vesicles. The authors need to calculate the number of exosomes produced, for example by the NTA method, and provide data normalized to the number of exosomes.

Figure 3A for Mitotracker red analysis is presented in low quality, which does not allow identification of the classic pattern of the mitochondrial net. The authors need to provide images at a higher magnification to demonstrate specific staining for Mitotracker red. The authors note that no significant differences were found between the raw mean intensity per cell in each of the modulations performed in SH-WT or SH-SWE cells. However, normalization of the fluorescence signal per neuron area revealed significant differences in Mitotracker intensity. These results may indicate not a change in the transmembrane mitochondrial potential, but a change in the area of cells. In this regard, the authors need to give the area of cells for each group on a separate graph. It should be noted that this method is associated with many nuances that can affect the final analysis, therefore, the authors need to provide a step-by-step protocol for measuring the cell area and normalizing the fluorescence intensity signal. More informatively, mitochondrial changes in cells would be described if the authors also estimated the total mitochondrial mass by the Mitotracker green dye.

Based on the data obtained, the authors figured out incorrect conclusions “Increased mitochondrial mass based on the enhanced intensity of Mitotracker red signal was observed by a mimic in either SH-WT or SH-SWE cells”. Using a ΔΨm-dependent dye, it is impossible to judge the mitochondrial mass. The authors should make a conclusion that corresponds to the experimental data.

Minor comments

Figure 1B shows low-quality transmission electron microscopy images for iNEU cells, it is impossible to distinguish round cup-shaped structures of exosomes, please provide the proper image.

How was the co-cultivation of cells carried out, it is desirable to give a graphical scheme of the experiment.

Author Response

C1- The authors make quantitative comparisons of total RNA levels ...

R1— We thank the reviewer for calling our attention for this issue and we completely agree with his/her considerations. When we mentioned the exosomal RNA and protein differences, we were considering that exosomes resulted from the same original number of cells 5x104 (either SH or iNEU) and that such variations could be derived from the elevated concentration of exosomes visible in TEM images from iNEU cells. At the time of the manuscript submission, we decided to not include the NTA histograms, considering that despite having performed the NTA analysis in SH cells (n=3), we only had the possibility to do the same assessment in one sample of either iNEU-PSEN or iNEU-WT cell-derived exosomes. Considering the request of the reviewer, and that it would be impossible to have more determinations in a short period of time, we chose to include such data after all, because NTA results are in line with those from TEM, RNA, and protein analysis, i.e., that an increased number of exosomes are released from iNEU cells. Considering the peak variation found in the histograms from the several samples, normalization to the number of exosomes results quite difficult. Nevertheless, once the elevated concentration of exosomes was observed in either iNEU-WT or iNEU-PSEN secretomes, we believe that differences in miRNA profiling are related with the presence of the AD mutation and not only with the density of vesicles. All these issues are now considered in the revised manuscript and the new representative NTA panels were included in the previous Figure 1 (please see Figure 1 NEW VERSION - panel D). The text now reads as follows:

Representative histograms by NTA analysis of exosomes from SH-WT and SH-SWE samples (n=3) and iNEU-WT and iNEU-PSEN secretome (n=1) are depicted in Figure 1D and are in line with the other described results indicating that an enhanced number of exosomes are possibly released by the iNEU cell lines, probably derived from the stressful conditions of their generation [40], and eventually aggravated by the PSEN mutation (1.7-fold elevation vs. respective WT; p < 0.05) (Supplementary Table S2). Differences in the biogenesis of exosomes should be explored in the future for a larger number of samples and extended to other iNEU cell lines derived from other mutations. To note that some variation in exosome size distribution profiles in the fresh collected samples was observed, though the most prevalent size in all samples was near 100 nm, the most usual in exosomes. We cannot however disregard that some aggregation processes may justify the peaks over 100 nm.

C2- Figure 3A for Mitotracker red analysis is presented in low quality, ....

R2- We understand reviewer concerns and we now provide images at higher magnification (Figure 3 NEW VERSION – panel A) to demonstrate specific staining for MitoTracker red, though the quality of the images is always better in the submitted TIF original images than in the word document version. We agree that, by normalizing the raw fluorescent intensity per neuron area, differences may be explained just by cell area. As requested by the reviewer we included the area of the cells for each group in the new Supplementary Table S4 in the revised manuscript. Though no significant differences were obtained, we found changes in cell area between the SH-WT and SH-SWE treated with the mock, the inhibitor, and the mimic. However, in the case of SH-WT cells the increase in the Mitotracker intensity seems to not derive from the cell area, but to be induced by the mimic, considering that the cell area in the mock-treated SH-WT cells (1037 µm2) are close to that of the same cells treated with the mimic (961 µm2). Therefore, though we cannot disregard the contribution of cell area differences to Mitotracker Red intensity increase by the mimic, namely in SH-SWE cells, it cannot be considered the only responsible factor for the findings achieved. These considerations were addressed in the revised highlighted manuscript (lines 477-488) and a detailed description for the cell area measurement to normalize the fluorescence intensity signal was included in the Materials and Methods Section (lines 380-385), as follows:

To note that, though no significant differences were obtained, we found changes in cell area between the SH-WT and SH-SWE treated with the mock, the inhibitor, and the mimic. However, in the case of SH-WT cells the increase in the Mitotracker intensity seems to not derive from the cell area, but to be induced by the mimic, considering that the cell area in the mock-treated SH-WT cells (1037 µm2) are close to that of the same cells treated with the mimic (961 µm2) (Supplementary Table S4). Therefore, though we cannot disregard the contribution of cell area differences to Mitotracker Red intensity increase by the mimic, namely in SH-SWE cells, it cannot be considered the only responsible factor for the findings achieved.

Cells were incubated for 30 min at 37°C with 500 nM of MitoTracker Red CMXRos®, according to manufactured instructions (Thermo Fisher Scientific), to stain viable mi-tochondria, and then fixed with 4% (w/v) paraformaldehyde [11]. Cell nuclei were stained with Hoechst 33258 dye. Images were acquired as aforementioned and total fluorescence intensity (FI) of the Mitotracker Red was assessed using ImageJ software. After all images were scaled, they were converted to black and white (B&W) images (Image>Colour Threshold). FI and cell area were automatically measured using (Ana-lyze>Analyze Particles) with the options “area” and “integrated intensity” selected from the menu “set measurements”. Then, the FI of MitoTracker Red was normalized using the calculated cell size (Supplementary Table S4). In total, over 320 cells were processed for each treatment group.

C3- Based on the data obtained, the authors figured out incorrect conclusions ...

R3- We thank the reviewer for calling our attention for the wrong use of the term “mitochondrial mass” associated to the Mitotracker green dye, which was now replaced by that of “transmembrane mitochondrial potential” in the revised version of the manuscript.

C4- Figure 1B shows...

R4- We provided better quality images to highlight the round morphology of the exosomes and the cup shape of surface-desiccated exosomes in high magnification. The new TEM images included in Figure 1 also evidence the increased number of exosomes released by iPSCs-derived neurons, either WT or PSEN, that are validated by the NTA representative histograms.

C5- How was the co-cultivation of cells...

R5- As requested by the reviewer we included in Figure 6 a graphical scheme elucidating how the co-cultivation of the cells was carried out.

We thank the reviewer comments that contributed to improve the scientific quality of our manuscript.

Reviewer 3 Report

The manuscript by Garcia et al demonstrates mutation-independent increased neuronal expression of miR-124, miR-125b and miR-21 in 2 cellular models of AD, and suggests that miR-124 levels are a potential therapeutic target to modulate neuroprotection.  Interestingly, some different responses to miR-124 modulation were observed in the 2 models indicting the importance of selecting the appropriate model to be used, and also translating their findings.  The findings of this study will be of great interest to the field, I just have some minor comments for the authors:

  1. Please clarify whether dopaminergic neurones (line 113) or hippocampal neurones (line 170) were generated from the iPSCs
  2. iPSCs were derived from one AD patient & one control, the authors should acknowledge the limitations of using just one disease & one control case
  3. The results section is often very repetitive of the methods (for example lines 378-386). Similarly avoid discussing the results in this section (for example lines 425-427). Please ensure that only the results are succinctly stated
  4. Figure 3A, what are the white arrows indicating?

Author Response

We thank the reviewer for considering our study of great interest to the AD field.

C1- Please clarify whether dopaminergic neurones (line 113) or hippocampal neurones (line 170) were generated from the iPSCs

R1 – We thank the reviewer for the pertinent question. The original protocol developed by Chambers and colleagues (2009) confirms the dopaminergic patterning. Further adaptations are from other studies that optimized conditions for midbrain dopaminergic neurons, such as Perrier et al (2004) (doi.org/10.1073/pnas.0404700101) or Kriks S et al (2011) (doi:10.1038/nature10648). In conformity, we now clarify in both lines of the manuscript that they are midbrain dopaminergic neurons.

C2- iPSCs were derived from one AD patient & one control, the authors should acknowledge the limitations of using just one disease & one control case

R2 - The authors recognize that this report is a pilot study concerning the limited models used to explore miR-124 role in the AD context. We addressed such limitations in the Results (lines 854-857 in the highlighted version of the manuscript) and in the Discussion sections of the revised manuscript (lines 896-898). Authors are aware that data need to be validated in other AD cell lines, including sporadic patients and different mutations.

C3- The results section is often very repetitive of the methods (for example lines 378-386). Similarly avoid discussing the results in this section (for example lines 425-427). Please ensure that only the results are succinctly stated.

R3 - The authors acknowledge the Reviewer concerns and removed repetitive experimental information and the discussion of data from the Results section, thus assuring a mere description of the results.

C4- Figure 3A, what are the white arrows indicating?

R4 – We thank the Reviewer for calling our attention for the missing information in the Figure 3 caption. The meaning of the white arrowheads is now explained in the main text and in the legend of such Figure. The white arrowheads indicate mitochondrial clamps in neurites.

Authors revised the English language and improved the writing of the manuscript.

Round 2

Reviewer 2 Report

The manuscript improved a lot after revision, the authors effectively responded to the reviewer comments presenting interesting and important results in an appropriate way with the appropriate methods and scientific background. I have no further suggestions and recommend the manuscript for publication.

Reviewer 3 Report

The authors have adequately addressed all the comments & the manuscript is now suitable for publication